# Detection of Aldehydes in Meat Products Based on Sulfonated Polystyrene Microspheres Modified with 2,4-Dinitrophenylhydrazine as Membrane-Protected Solid-Phase-Extraction Adsorbents

**DOI:** 10.3390/foods15010101

**Published:** 2025-12-29

**Authors:** Siyi Wang, Shibing Zhang, Min Fu, Siying Lu, Qi Zhao

**Affiliations:** 1State Key Laboratory of Marine Food Processing and Safety Control, Dalian Polytechnic University, Dalian 116034, China; www011478@163.com (S.W.); 13470306550@163.com (S.Z.); 15905415700@163.com (M.F.); 15895679280@163.com (S.L.); 2National Engineering Research Center of Seafood, School of Food Science and Technology, Dalian Polytechnic University, Dalian 116034, China

**Keywords:** membrane-protected micro-solid-phase extraction, sulfonated polystyrene microspheres aldehydes, meat samples, HPLC-MS/MS

## Abstract

Sulfonated polystyrene (sPS) microspheres modified with 2,4-dinitrophenylhydrazine were prepared and used as adsorbents in membrane-protected solid-phase microextraction (μ-SPE) to extract aldehydes from different meat samples during thermal processing. Fourier transform infrared spectroscopy indicated the successful modification of sulfonic acid groups on the surface of PS microspheres. Under the optimized adsorption and analytical conditions, aldehydes can be effectively extracted from meat samples with excellent regression linearities (0.9921–0.9993), extremely low limits of detection (0.010–0.621 ng g^−1^), satisfactory recovery rates (72–107%), and relatively accurate precisions (1–11%). Moreover, the proposed method also eliminated interferences using a polytetrafluoroethylene (PTFE) membrane in the µ-SPE device and could realize extraction, separation, enrichment, and derivatization integration in one step. Therefore, the developed method provides a promising alternative to the reported methods.

## 1. Introduction

As widespread naturally ubiquitous carbonyl compounds, aldehydes are found in foodstuffs produced by Maillard, caramelization, and lipid peroxidation reactions during food processing, especially frying and fermentation processes [1]. The deterioration of taste, flavor, odor, color, texture, and appearance and decreases in nutritional value occur during the formation of aldehydes. In addition, aldehydes are considered as compounds to which exposure increases the risk of a lot of diseases. For example, formaldehyde was classified as class I carcinogen by the International Agency for Research on Cancer in 2005 [2]. Other aldehydes such as acetaldehyde and malondialdehyde have also been reported to be carcinogenic [3]. Since some aldehydes are unavoidable in fried meat, the sensitive determination of the aldehyde content in fried meat is essential for meat food quality maintenance.

A variety of methods such as gas chromatography (GC) [4], high-performance liquid chromatography (HPLC) [5], electroanalytics [6], capillary electrophoresis [7], spectrophotometry [8], mass spectrometry [9], and microfluidic paper-based analytical devices [10] have been developed for aldehyde analysis. Nonetheless, due to the diversity of molecular aldehydes, GC and HPLC are the most used technologies based on their ability to analyze multiple analytes. However, for GC with different detectors like FID and MS, poor separation ability and low precisions have been observed [11]. Hence, the derivatization of carbonyls with 2,4-dinitrophenylhydrazine (DNPH), followed with LC-MS/MS, is the most selective and sensitive method for quantification analysis of aldehydes.

The detection of aldehyde compounds is challenging due to their low volatility and high reactivity due to their polar carbonyls. Currenlt, the determination of aldehydes is based on the reaction of the carbonyl group with DNPH to form the corresponding hydrazone, followed by liquid–liquid extraction [12] or solid-phase extraction (SPE) [7,13]. Among these, SPE is a common sample preparation method used to remove interfering compounds. So far, a variety of adsorbents, including polydimethylsiloxane-divinylbenzene (PDMS-DVB), solid-phase microextraction (SPME) fibers [14], SDS-alumina coated magnetic nanoparticles [15], ZnO nanorod coated fibers [16], and magnetic strong cation-exchange resins [17], have been used as SPE adsorbents for the enrichment and purification of aldehydes in different samples. Among these, strong cation-exchange sulfonated polystyrene (sPS) microspheres charged with DNPH via ionic and hydrophobic interactions are of particular interest, with the advantages of both fulfilling the extraction and derivatization synchronously and reducing the amount of the derivatization reagent required [17].

However, when applying SPE to meat samples, additional steps, such as centrifugation, protein precipitation, and filtration, are generally required to avoid blockage of the SPE columns [18]. As an alternative, miniaturized sorbent phase-based extraction methods, including SPME [13], magnetic SPE (MSPE) [19] and the membrane-protected solid-phase microextraction technique (μ-SPE) [20,21], have been developed and applied for aldehyde analysis in food samples. Compared to SPME and MSPE with direct immersion, μ-SPE can extract targets from complex matrices without additional cleaning steps by avoiding contact between the adsorbents and the matrix components based on the membrane protection effect [22]. So far, limited studies have examined the application of μ-SPE for the extraction of aldehydes from meat samples.

The sPS microspheres were prepared by bestowing sulfonic acid groups (-SO_3_H) upon the surface of PS microspheres. The prepared sPS microspheres were characterized by scanning electron microscopy (SEM), energy spectrum analysis (EDS), Fourier transform infrared spectroscopy (FTIR), and thermogravimetric analysis (TGA). The sulfonic acid groups on the surface of the sPS microspheres served as anchor sites for the immobilization of DNPH through ionic interactions [23]. Then, the stable hydrazone conjugates formed between the hydrazine moieties in DNPH and the carbonyl groups in aldehydes were captured by the sPS microspheres via affinity addition interaction. Consequently, the DNPH-modified sPS microspheres could efficiently extract aldehyde compounds from complex matrices with minimal interference from extraneous species [24]. In addition, the performance of the sPS microspheres used as μ-SPE sorbents for the extraction of aldehydes from meat samples was also investigated.

## 2. Materials and Methods

### 2.1. Chemicals and Reagents

HPLC-grade acetonitrile and formic acid were provided by Merck (Darmstadt, Germany). Polyvinyl alcohol (PVA) and 2,4-dinitrophenylhydrazine (DNPH) were purchased from Aladdin Reagent (Shanghai, China). Other reagents of analytical grade such as styrene (St), divinylbenzene (DVB), azobisisobutyronitrile (AIBN), 1,2-dichloroethane, and methanol were provided by Damao Chemical Reagent Factory (Tianjin, China). A polytetrafluoroethylene membrane (PTFE, 0.45 μm) was purchased from Shanghai Xingya Purification Material Factory (Shanghai, China).

Standards of aldehydes, including crotonaldehyde, butanal, pentanal, trans-2-hexenal, hexanal, trans, trans-2,4-heptadienal, heptanal, trans, trans-2,4-octadienal, octanal, nonanal, and trans, trans-2,4-decadienal were purchased from ANPEL Laboratory Technologies (Shanghai, China). First, 10 mg of each aldehyde standard mentioned above was weighed, dissolved in ethanol/water (1:1, *v*/*v*) to a constant volume of 10 mL, and stored at −20 °C. A stock mixture of aldehyde standards were prepared at a concentration of 20 μg/mL using ethanol/water (1:1 *v*/*v*) and stored at 4 °C for subsequent use.

### 2.2. Instruments

The determination of target substances was performed on an LC-10AD/4000QTRAP tandem mass spectrometer (Shimadzu Co., AB Sciex, Kyoto, Japan). The detection process and data analysis were completed using Analyst software equipped with Multiquant 2.1 software. The morphology was recorded by means of scanning electron microscopy (SEM) with a TM 3000 (Hitachi, Tokyo, Japan). The main elements of the adsorbent were detected by an energy dispersive spectrometer (EDS) (Oxford Instruments, X-MaxN, Abingdon-on-Thames, UK). In addition, the functional groups were verified by means of Fourier transform infrared (FT-IR) spectroscopy (Perkin Elmer, Norwalk, CT, USA).

### 2.3. The Preparation of PS Microspheres

The preparation of PS microspheres was carried out according to emulsion polymerization described by Gelir et al. [25], with slight modifications. The reaction process is shown in Figure 1. In short, the initiator AIBN (0.3 g) was dissolved in a mixed solution of St (10.0 mL) and DVB (2.0 mL) to prepare a premixed solution. Then, 100.0 mL of PVA aqueous solution (5.0 g L^−1^) was added to the reactor (250 mL), and the mixed solution was stirred at 450 rpm and protected with nitrogen. After the temperature reached 90 °C, the premix was added to the reactor. Finally, after stirring for 6 h, the obtained polymer particles were washed with ethanol and dried at 60 °C to a constant weight.

### 2.4. The sPS Microspheres

The sPS microspheres were obtained according to the method of Zhou et al. [26]. First, 3.0 g of dried PS microspheres were immersed into 20.0 mL of 1,2-dichloroethane in a water bath at 60 °C. Then, 30 mL of concentrated sulfuric acid was added to the system and reacted at 80 °C for 2 h. Finally, the mixture was washed with water to neutrality and dried at 60 °C.

### 2.5. The Preparation of DNPH Modified sPS Microspheres

After the DNPH was recrystallized twice with ACN, the sPS microspheres were immersed in saturated DNPH ethanol aqueous solution (10%, *v*/*v*) for 12 h. Finally, the sPS microspheres obtained were washed with water until the washing solvent was colorless. Then, DNPH-modified sPS microspheres were dried at 40 °C and stored in a sealed brown flask.

### 2.6. The Preparation of the Membrane-Protected μ-SPE Device

The PTFE membrane with a pore size of 0.45 μm was folded in half and the two edges were heat sealed with a heat sealer. Then, 29.0 mg of dried DNPH-modified sPS microspheres were placed into the self-made plastic sleeve. Finally, the edge of the opening was heat sealed and the final size of the μ-SPE device was 2 cm × 2 cm.

### 2.7. Membrane Protected Solid Phase Extraction by μ-SPE Device

First, a 1.0 g meat sample, 0.5 mL of acetonitrile, and 4.5 mL of 0.02 mol L^−1^ phosphate buffer (pH = 6.0) solution were placed into a centrifuge tube and the mixture was extracted under ultrasonic treatment for 15 min. The ultrasonic operation parameters are as follows: frequency, 20 kilohertz; rated power, 450 watts; actual output power set at 60% (270 watts); pulse mode operation (3 s on, 2 s off); total processing time, 15 min. Then, the μ-SPE device was added and the aldehydes were adsorbed and derivatized at 70 °C with vigorous magnetic stirring at 800 rpm. After extracting for 15 min, the μ-SPE device was taken out and wiped with filter paper. Then, the μ-SPE device adsorbed with aldehydes was washed with distilled water 3 times and eluted with 1.0 mL of 5% ammoniated methanol solution 4 times, with 1 min of ultrasonic assistance each time. Finally, the residue was re-dissolved with 4.0 mL of the mobile phase and filtered using a 0.22 μm filtration membrane for further HPLC-MS/MS analysis.

### 2.8. HPLC-MS/MS Conditions

The separation of aldehydes was carried out on a Hypurity C18 column (150 mm × 2.1 mm, 5 μm; Thermo Scientific, Waltham, MA, USA) with the mobile phase of solvent A (0.1% formic acid water solution) and solvent B (acetonitrile). The injection volume was 20 μL and the flow rate was 0.2 mL min^−1^, with a column temperature of 30 °C. The linear gradient elution procedure was as follows: 0–2.0 min, 40% A; 2.0–8.0 min, 40–15% A; 8.0–10.0 min, 15% A; 10.0–11.0 min, 15–5% A; 11.0–13.0 min, 5% A; 13.0–14.0 min, 5–60% A; 14.0–15.0 min, 60% A. ESI was carried out in negative mode, and the optimized working parameters were as follows: ion source temperature, 425 °C; ion spraying voltage, 4500 V; ion source gas 1 (nitrogen, purity 99.9%), 45 psi; ion source gas 2 (nitrogen, purity 99.9%), 40 psi. The multiple reaction monitoring (MRM) mode was used for MS/MS analysis. The MRM condition was automatically optimized by injecting 1000 ng mL^−1^ aldehyde standards into the spectrometer and the collision energy, the exit potential of the collision cell, and the declustering potential of the diagnostic ion were obtained. The parameters are shown in Table 1.

### 2.9. Statistical and Analysis

The experiment was conducted three times, and the results were presented as the mean ± standard deviation. The mean differences between the two independent samples were compared through *t*-tests. The t-value was calculated using the following formula. The corresponding *p*-value was found based on the t-distribution table. If *p* < 0.05, it was concluded that the mean differences between the two groups were significant.
$$t=\frac{{\bar{x}}_{1}-{\bar{x}}_{2}}{\sqrt{\frac{s_{1}^{2}}{n_{1}}+\frac{s_{2}^{2}}{n_{2}}}}$$ where the means of the two sets of data are represented by *x*_1_ and *x*_2_, *s*_1_ and *s*_2_ represent the standard deviations of the two sets of data, and *n*_1_ and *n*_2_ represent the number of data points in the two sets.

## 3. Results and Discussion

### 3.1. The Characterization of sPS Microspheres

The SEM images of the PS microspheres and sPS microspheres are shown in Figure 2a,b, and the result indicate that the PS microspheres have a highly uniform spherical shape with a diameter of about 50 μm, and the morphology was not destroyed after the sulfonation procedure. Furthermore, to identify the chemical composition of sPS microspheres, the EDS spectra of sPS are illustrated in Figure 2c–g. The main elements of the microspheres were C, O, and S. The uniform S elements (Figure 2g) were loaded on the surface of PS microspheres, indicating that the sulfonation reaction on the surface of PS microspheres was successfully carried out.

The FT-IR spectra of PS microspheres and sPS microspheres are shown in Figure 3a. The absorption peaks of PS microspheres and sPS microspheres at 1600 cm^−1^, 1492 cm^−1^, and 1451 cm^−1^ are due to the tensile vibration of the benzene ring skeleton structure [27]. The absorption bands of PS microspheres and sPS microspheres near 755 cm^−1^ and 697 cm^−1^ correspond to the characteristic absorption peaks of single-group substitution on the benzene ring, while the absorption bands at 3024 cm^−1^ and 2914 cm^−1^ are attributed to the C-H stretching vibration. These peaks are consistent with the standard spectra of polystyrene [28]. In addition, the absorption peaks of sPS microspheres at 1180 cm^−1^ and 1027 cm^−1^ are due to the stretching vibration of S=O, indicating successful modification of the surface of PS microspheres by the sulfonic acid group.

The thermal stability of sPS microspheres was characterized and the thermogravimetry (TG) and derivative thermogravimetry (DTG) are shown in Figure 3b. The mass of the samples decreased drastically in the temperature range of 350–450 °C, and the weight loss was due to the decomposition of the PS [26]. Hence, the prepared sPS microspheres had good thermal stability in the range of experiment temperature, which could meet the experimental requirements.

### 3.2. Optimizing the Extraction Conditions

In the experiment, the extraction recovery rates of aldehydes were affected by various factors, such as the conditions of the membrane-protected μ-SPE device and the extraction and elution conditions. Hence, the effect of different conditions on the recovery rates was evaluated. No matter which parameter was optimized, the other parameters remained at their optimal values. The samples for all optimization experiments were spiked chicken samples (50 ng g^−1^).

#### 3.2.1. The Effects of the μ-SPE Device

The effects and interactions of three factors, including the number of μ-SPE devices, the surface area of the μ-SPE device, and the amount of adsorbent in the μ-SPE device, on the recovery rates were examined using Box–Behnken matrices based on 17 sets of data, and the response values are shown in Appendix A. In addition, various parameters of the designed experiments are shown in Appendix A. Analysis of variance showed that the *p* values were all less than 0.05, indicating that the model terms were statistically significant (i.e., a probability < 5% indicated that the observed effects were due to random variation). The lack-of-fit F-value was not significant (*p* > 0.05), confirming that the model adequately describes the data without systematic bias. These results indicate that the model was fitted well. Finally, the amount of adsorbent added was set at 28.78 mg, the optimized number of μ-SPE devices was 1, and the single-sided surface area of the device was 4 cm^2^ (2 cm × 2 cm). In order to facilitate the experimental operation, the added amount of adsorbent was determined to be 29.0 mg. The effect of each parameter on the recovery of hexanal are shown in Figure 4.

#### 3.2.2. Extraction Conditions

In order to achieve more efficient detection, the effects of different extraction methods on recovery rates were evaluated in our experiment. As shown in Figure 5a, compared with stirring extraction, with the assistance of ultrasonication, the range of individual mean recovery rates observed among different aldehydes studied was increased to 71 ± 3–97 ± 5%. Hence, ultrasound-assisted extraction was selected as the extraction method and the effect of ultrasonic extraction time (5 min, 10 min, 15 min, 30 min) on the recovery rates was further investigated (Figure 5b). The recovery rates of aldehydes increased with extraction time. When the ultrasonic extraction time reached 15 min, the recovery rates of aldehydes were 69 ± 2–96 ± 6%. Then, with the further extension of the ultrasonic extraction time, no significant change was reflected in the recovery rates. In the follow-up experiments, 15 min was used as the optimal ultrasonic extraction condition.

#### 3.2.3. Elution Conditions

The pH screening experiment (pH 4.0–8.0) evaluated the extraction buffer pH. At pH 4.0, recovery rates ranged from 45 ± 3 to 68 ± 3%; as pH increased from 4.0 to 6.0, recovery rates increased from 45 ± 1 to 68 ± 3% to 82 ± 5–97 ± 3%; when pH further increased from 6.0 to 8.0, recovery rates decreased from 82 ± 2 to 97 ± 4% to 52–71%. Therefore, pH 6.0 was optimal for extraction. Notably, the alkaline eluent (ammoniated methanol, pH~11) is used specifically during the elution step to disrupt ionic bonds between DNPH derivatives and sulfonic acid groups, which is mechanistically distinct from the extraction pH optimization. To choose a suitable desorption solvent, the effect of elution solvent on the recovery rates was investigated using different concentrations of ammoniated methanol solutions (5%, 10%, 15% and 20%). When 5% ammoniated methanol solution was used as the eluent, the recovery rates of aldehydes were satisfactory (73.66–106.94%), being significantly better than for the other elution conditions (Figure 5c).

At the same time, the effect of elution solution volume (1.0 mL, 2 × 1.0 mL, 3 × 1.0 mL, 4 × 1.0 mL, 5 × 1.0 mL) on the recovery rates was also investigated (Figure 5d). When the number of elutions increased from 1 to 4, the recovery rates increased from 35–64% to 73–113%. When further increasing the number of elutions, insignificant changes were observed in the recovery rates. Therefore, 4 × 1.0 mL of 5% ammoniated methanol solution was the optimal elution condition in this study.

### 3.3. Method Verification

Method verification was carried out under the optimal extraction conditions, and the matrix effect, linear range, LODs and LOQs of the method were evaluated, with the results being shown in Table 2. Excellent linearities in the range of 0.1–1000 ng mL^−1^ were attained, and good regression coefficients were obtained between 0.9921 and 0.9993. LODs and LOQs were determined to be 0.010–0.621 ng g^−1^ and 0.034–2.069 ng g^−1^ based on signal-to-noise (S/N) ratios of 3:1 and 10:1, respectively.

The matrix effect caused by co-extractives of complex samples was a noteworthy problem, owing to it greatly damaging the accuracy of LC–MS/MS quantification at trace levels [29]. The enhancement value of the matrix effect of this method was between 2% and 16%, and the inhibition value was between 2% and 9%. Hence, the matrix effect of this method was weak (<20%). In addition, there were no significant differences in the matrix effects for different meat samples, indicating that the method was suitable for the analysis of different meat samples.

The precision of the method was determined by evaluating intra-day (six samples in one day) RSDs and inter-day (samples for six consecutive days) RSDs (Appendix A), which were 1–10% and 1–11%, respectively. The AOAC guidelines stipulate that when the analyte concentration is between 1 μg g^−1^ and 10 ng g^−1^, a method has good stability, while it has good reproducibility when the intra-day and inter-day precision values are less than 15% [30]. Hence, this method had good stability and reproducibility. Moreover, the range of individual aldehyde recovery rates at three spiking concentrations was 72 ± 5% (trans, trans-2,4-decadienal) to 106 ± 4% (hexanal). Each value represents the mean recovery for a specific aldehyde compound across all meat matrices tested. According to the U.S. Food and Drug Administration, recovery rates should be in the range of 70–120% when the concentration of analyte is at the 10 ppb level. So, this experimental method enables the accurate quantification of aldehydes. In addition, the specific chromatographic peaks for the quantitative analysis of 50 ng mL^−1^ aldehydes in blank chicken samples are shown in Figure 6.

### 3.4. Study of Adaptability

To demonstrate the applicability and stability of the proposed method in the preparation of complex samples, we applied it to the analysis of aldehyde substances in different grilled meat products. As shown in Table 3, the initial aldehyde concentration in the meat samples significantly increased after intense heat treatment (under high-temperature grilling conditions), and the specific data are presented in Appendix A. Specifically, before and after barbecuing, the content of each aldehyde in the samples was changed as follows: in pork samples, the content of valerolactone increased from 71.12 ± 3.15 ng g^−1^ to 114.08 ± 2.02 ng g^−1^, an increase of 60%; that of hexalactone increased from 41.53 ± 1.58 ng g^−1^ to 92.43 ± 5.09 ng g^−1^, an increase of 123%; that of heptalactone increased from 4.49 ± 0.20 ng g^−1^ to 56.89 ± 2.67 ng g^−1^, an increase of 1167%; that of octalactone increased from 10.06 ± 0.58 ng g^−1^ to 54.21 ± 2.24 ng g^−1^, an increase of 439%; and that of nonalactone increased from 78.19 ± 4.42 ng g^−1^ to 122.16 ± 4.57 ng g^−1^, an increase of 56%. In chicken samples, the content of valerolactone increased from 23.09 ± 0.40 ng g^−1^ to 68.10 ± 3.24 ng g^−1^, an increase of 195%; that of hexalactone increased from 15.56 ± 1.17 ng g^−1^ to 65.31 ± 1.01 ng g^−1^, an increase of 320%; and that of nonalactone increased from 32.05 ± 2.81 ng g^−1^ to 75.42 ± 2.71 ng g^−1^, an increase of 135%. In fish samples, the content of hexalactone increased from 13.81 ± 0.93 ng g^−1^ to 66.94 ± 5.81 ng g^−1^, an increase of 385%; that of heptalactone increased from 5.03 ± 0.59 ng g^−1^ to 52.06 ± 7.17 ng g^−1^, an increase of 935%; and that of octalactone increased from 4.17 ± 0.70 ng g^−1^ to 50.74 ± 3.80 ng g^−1^, an increase of 1117%. In beef samples, the content of valerolactone increased from 109.56 ± 1.03 ng g^−1^ to 152.93 ± 5.02 ng g^−1^, an increase of 40%; that of heptalactone increased from 43.97 ± 2.20 ng g^−1^ to 88.48 ± 2.29 ng g^−1^, an increase of 101%; and that of octalactone increased from 48.95 ± 5.09 ng g^−1^ to 91.89 ± 5.46 ng g^−1^, an increase of 88%. In mutton samples, the content of valerolactone increased from 26.77 ± 1.33 ng g^−1^ to 69.77 ± 1.05 ng g^−1^, an increase of 161%; and that of hexalactone increased from 16.76 ± 1.15 ng g^−1^ to 70.44 ± 7.03 ng g^−1^, an increase of 320%. These results clearly indicate that with the increase in heating time and temperature, the concentration of aldehydes in the meat samples significantly accumulates, and all changes are statistically significant (*p* < 0.05).

In addition, to further evaluate the accuracy of this method, the recovery rates of aldehydes in six kinds of spiked meat (50 ng g^−1^) were evaluated and the results are shown in Appendix A. The recovery rates of aldehydes in the six spiked grilled samples were 78–110%, with an RSD of less than 14%. These results show that the membrane-protected μ-SPE method was feasible and stable, so it could be applied to the analysis of aldehydes in complex food samples.

### 3.5. Comparison of Established Methods with the Reported Method

A comparison of this method with other existing techniques for measuring the content of aldehydes are summarized in Table 4, including analytical methods, adsorbents, and analytical parameters. The recovery rates and precisions of our method were comparable to other methods in the related articles. Nevertheless, the LODs of our method (0.010–0.621 ng g^−1^) were significantly superior to the existing methods based on the high sensitivity of triple quadrupole mass spectrometry in quantification. In addition, compared with commercial cartridges, commercial adsorbents, and SPME needles, the sPS microspheres have the advantages of easier preparation, lower consumption cost, and excellent adsorptive performance through functional modification. Moreover, the proposed method also eliminated interferences using a PTFE membrane in the µ-SPE device and could realize extraction, separation, enrichment, and derivatization integration in one step. Therefore, the developed method was considered to provide a promising and alternative method for the analysis of aldehydes in the future.

## 4. Conclusions

In this study, sPS microspheres modified with DNPH were prepared, and Fourier transform infrared spectroscopy indicated the successful modification of sulfonic acid groups on the PS microspheres’ surface. Then, sPS microspheres were used as the adsorbent in μ-SPE to analyze aldehydes for different meat samples during thermal processing. The optimized method demonstrated excellent regression linearities (0.9921–0.9993), extremely low limits of detection (0.010–0.621 ng g^−1^), and satisfactory recovery rates (72–106%), with relative standard deviations (RSDs) ranging from 1% to 11%. The proposed method eliminated interferences by using a PTFE membrane in the μ-SPE device and realized the integration of extraction, separation, enrichment, and derivatization in one step. This integrated approach not only simplified the sample preparation process but also enhanced the efficiency and sensitivity of aldehyde detection. The method was validated through the analysis of various meat samples, including pork, beef, mutton, chicken, chicken wings, and fish, demonstrating its applicability and stability in complex food matrices. Compared with other existing techniques, such as commercial adsorbents and SPME needles, the sPS microspheres offer easier preparation, lower consumption costs, and excellent adsorptive performance through functional modification. In summary, this study not only enhances the scientific depth of aldehyde detection but also provides a practical and reliable method for food quality and safety analysis.

## Figures and Tables

**Figure 1 foods-15-00101-f001:**
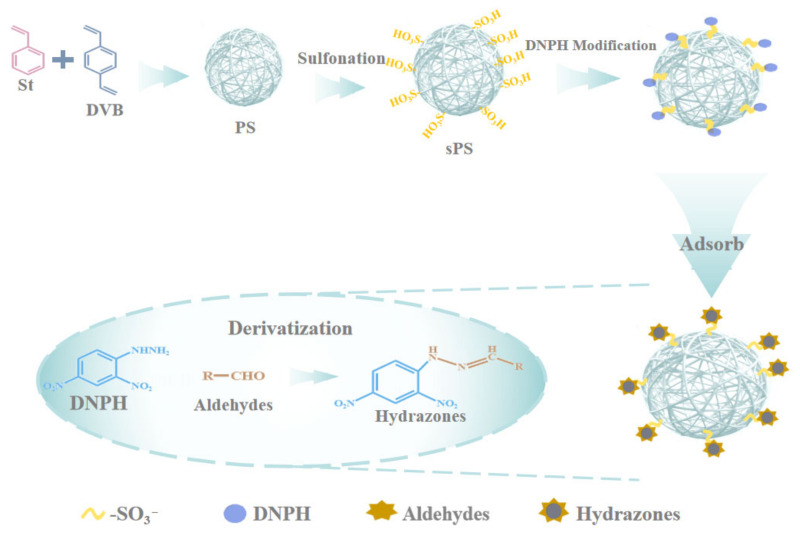
The schematic diagram for the preparation of sPS particles modified with DNPH.

**Figure 2 foods-15-00101-f002:**
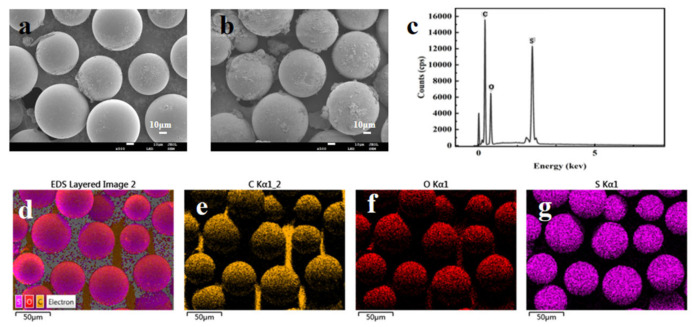
The SEM images of PS (**a**) and sPS microspheres (**b**); the EDS spectra of sPS microspheres (**c**); the EDS mapping images for sPS microspheres (**d**); the EDS mapping images for C (**e**), O (**f**)**,** and S (**g**) in sPS microspheres.

**Figure 3 foods-15-00101-f003:**
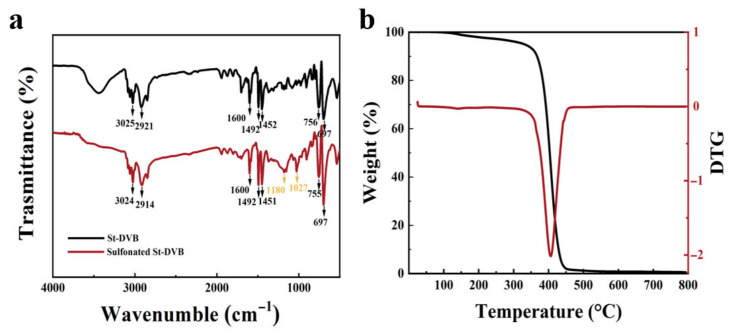
FTIR (**a**) and TG and DTG curves (**b**) of the sPS microspheres.

**Figure 4 foods-15-00101-f004:**
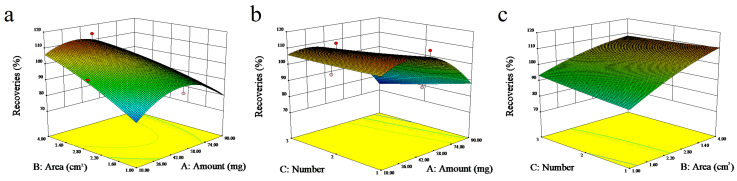
The effect of the adsorbent amount and the μ-SPE surface area (the number of μ-SPE was 1) (**a**), the adsorbent amount and the μ-SPE number (the surface area of μ-SPE was 4 cm^2^) (**b**) and the surface area and the μ-SPE number (the amount of adsorbent was 28.78 mg) (**c**) on the recovery of hexanal by the 3D response surface.

**Figure 5 foods-15-00101-f005:**
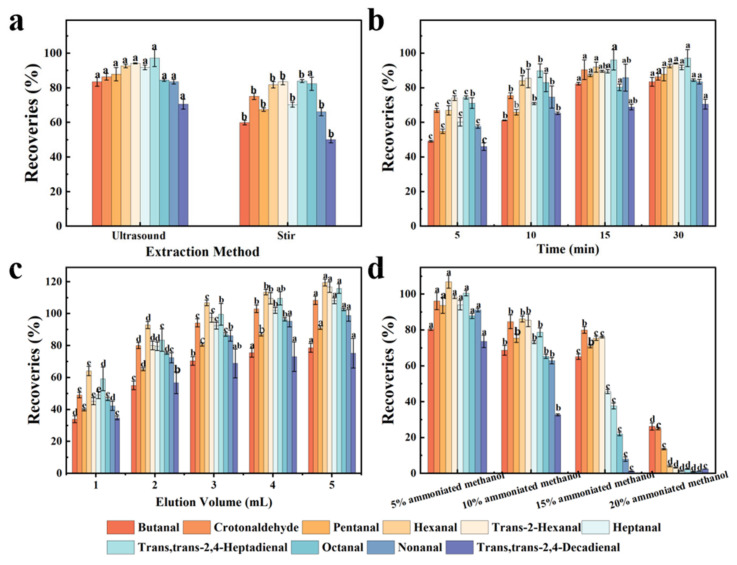
Effect of extraction method (**a**), time (**b**), elution solvent (**c**) and elution solvent volume (**d**) on the recoveries. Values of samples with different lower-case letters (a–e) are significantly different at *p* < 0.05.

**Figure 6 foods-15-00101-f006:**
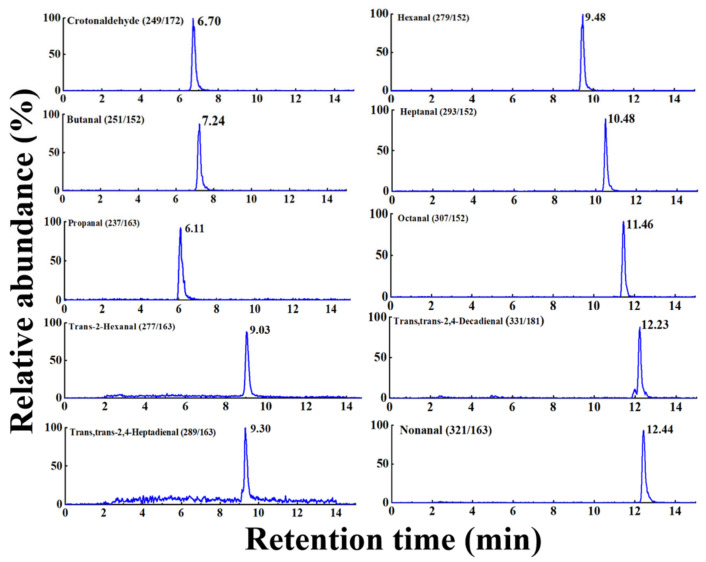
The specific extracted ion chromatograms for aldehydes in spiked chicken samples (50 ng mL^−1^).

**Table 1 foods-15-00101-t001:** The optimized mass spectrometry parameters for the aldehydes.

Analytes	RT (min)	Precursor Ion (*m*/*z*)	Product Ion (*m*/*z*)	CE (eV)	DP (V)	CXP (V)
Propanal	6.11	237	163	16	20	1
Crotonaldehyde	6.70	249	172	20	35	5
Butanal	7.24	251	152	22	25	1
Pentanal	8.34	265	152	26	35	25
Trans-2-Hexanal	9.03	277	163	20	25	1
Trans, trans-2,4-Heptadienal	9.30	289	163	16	15	7
Hexanal	9.48	279	152	28	35	5
Heptanal	10.48	293	152	28	25	1
Octanal	11.46	307	152	30	30	11
Trans, trans-2,4-Decadienal	12.23	331	181	32	5	31
Nonanal	12.44	321	163	20	25	9

Note: RT, retention time; CE, collision energy; DP, declustering potential; CXP, collision cell exit potential.

**Table 2 foods-15-00101-t002:** The matrix effect, linear ranges, regression equations, LODs, and LOQs of aldehydes.

Aldehydes	Matrix Effect (%)	Linear Equation	R^2^	Linear Ranges (ng mL^−1^)	LODs (ng g^−1^)	LOQs (ng g^−1^)
Crotonaldehyde	9	y = 1880.9x + 23,704.1	0.9966	1.0–1000.0	0.011	0.038
Butanal	7	y = 2977.9x + 47,609.1	0.9953	4.0–1000.0	0.021	0.071
Pentanal	6	y = 4868.1x + 166.4	0.9993	0.4–400.0	0.044	0.123
Trans-2-Hexanal	5	y = 2725.0x + 5402.0	0.9982	1.0–200.0	0.163	0.788
Trans,trans-2,4-Heptadienal	−9	y = 763.4x + 5497.2	0.9952	1.0–400.0	0.621	2.069
Hexanal	−5	y = 4213.3x + 20,679.3	0.9966	0.4–400.0	0.032	0.086
Heptanal	2	y = 3601.5x + 20,346.5	0.9946	0.2–400.0	0.010	0.034
Octanal	−2	y = 3071.0x + 18,135.0	0.9948	0.1–400.0	0.011	0.037
Trans,trans-2,4-Decadienal	16	y = 1166.2x + 7446.6	0.9921	0.4–400.0	0.054	0.148
Nonanal	6	y = 2874.6x + 3400.2	0.9990	0.2–200.0	0.025	0.082

**Table 3 foods-15-00101-t003:** Aldehyde content detected in six commercial meat products (ng g^−1^).

Aldehydes	Complex Sample					
	Pork	Beef	Mutton	Chicken	Chicken Wings	Fish
Crotonaldehyde	n.d.	n.d.	n.d.	n.d.	n.d.	n.d.
Butanal	n.d.	3.71 ± 0.57	n.d.	n.d.	16.24 ± 1.19	n.d.
Pentanal	71.12 ± 3.15	109.56 ± 1.03	26.77 ± 1.33	23.09 ± 0.40	164.64 ± 6.79	82.93 ± 2.68
Trans-2-Hexanal	n.d.	1.44 ± 0.30	n.d.	n.d.	1.70 ± 0.22	n.d.
Trans,trans-2,4-Heptadienal	n.d.	20.17 ± 1.93	n.d.	n.d.	n.d.	n.d.
Hexanal	41.53 ± 1.58	311.23 ± 4.58	16.76 ± 1.15	15.56 ± 1.17	81.23 ± 1.00	13.81 ± 0.93
Heptanal	4.49 ± 0.20	43.97 ± 2.20	n.d.	n.d.	7.26 ± 0.21	5.03 ± 0.59
Octanal	10.06 ± 0.58	48.95 ± 5.09	7.74 ± 0.98	n.d.	16.37 ± 4.92	4.17 ± 0.70
Trans,trans-2,4-Decadienal	25.33 ± 3.60	92.94 ± 9.57	n.d.	n.d.	109.83 ± 4.89	1.10 ± 0.19
Nonanal	78.19 ± 4.42	130.48 ± 8.07	128.17 ± 5.81	32.05 ± 2.81	79.25 ± 9.02	57.82 ± 7.72

**Table 4 foods-15-00101-t004:** Comparison of the current method with other extraction methods in aldehyde analysis.

Sample Preparation Method	SPE Adsorbent	Laboratory Preparation	LODs (ng g^−1^)	Recovery Rates (%)	RSD (%)	References
HS-SPME-GC/MS	Commercial PDMS-DVB SPME fibers	No	100–500	50–100	≤13	[14]
µ-SPE-LC/ESI/MS	Commercial Telos™ ENV PS co-polymer	No	90–300	95–99	6–9	[21]
µ-SPE-HPLC/UV	Commercial silica-based C_2_, C_8_, C_18_, DVB-MAA, DVB-PEI, DVB-NEP	No	70–150	84–106	7–12	[7]
HS-SPME-GC/FID	ZnO nanorod coated fiber	Yes	13–37	71–129	8–10	[16]
MSPE-HPLC/UV	SDS-alumina coated magnetic nanoparticles	Yes	4900–21,400	82–115	≤10	[19]
µ-SPE-HPLC/MS/MS	sPS microspheres	Yes	0.010–0.621	72–106	1–11	Our method

## Data Availability

The original contributions presented in the study are included in the article and Appendix A; further inquiries can be directed to the corresponding authors.

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
