# Peer review of "Detection of Aldehydes in Meat Products Based on Sulfonated Polystyrene Microspheres Modified with 2,4-Dinitrophenylhydrazine as Membrane-Protected Solid-Phase-Extraction Adsorbents"

_foods, 2025, doi:10.3390/foods15010101_

Round 1
Reviewer 1 Report
Comments and Suggestions for Authors
In general, the article is of interest from the point of view of expanding the possibilities of meat sample preparation and improving the characteristics of determining aldehydes. However, there are some questions and comments.
Common comments
The introduction doesn't entirely clarify the novelty of this work. Both DNPH-modified sulfonated polystyrene microspheres and μ-SPE have been used before. What's new is their combination or application to meat analysis? Or the simultaneous extraction and derivatization of aldehydes? Moreover it would be helpful to articulate the purpose of the work.
Authors often use abbreviations without introducing them (St-DVB, PTFE, PDMS-DVB, etc.). These are understandable to a narrow circle of specialists, but still require prior explanation.
A more detailed explanation of the chemical mechanism of modification of spheres is desirable.
The authors used three factors (the number of μ-SPE devices, the surface area of the μ-SPE device, and the amount of adsorbent in the μ-SPE) to plan sorption conditions, but they are largely correlated with each other. However, other extraction conditions, such as pH, time, temperature are not studied.
The authors investigate elution at different concentrations of ammoniated methanol solution, but why was this solution chosen as the eluent?
When comparing the Established Method with the Reported Methods, various detection methods are compared, while the authors propose a sample preparation method. Therefore, the advantages may be related to the detection method. It makes sense to compare sample preparation methods.
The authors use an excessive number of digits in the data. For example, 97.17 ± 4.91% should be represented as 97 ± 5. This also applies to Tables 2-6.
Special comments
Abstract
Line 15, “to analysis” – not for analysis, but for extraction/pre-concentration
Line 21, «limit of detection (0.010-0.621 ng/g)»: It is not clear what method of determination.
Line 22, «precisions (1.05-10.63%)»: What value is represented? Sr? Or some other?
2.1. Chemicals and Reagents
Why are DNPH from two different companies used?
Line 96, «The mixed standard stock solution of aldehyde»: aldehydes?
2.4. The Preparation of St-DVB Microspheres
Line 109, «The reaction process was shown in Figure 1»: This is more of a schema. The figure could be presented in a more "chemical" and informative manner. Although this is not necessary, as it is not new.
2.7. Membrane Protected Solid Phase Extraction by μ-SPE Device
Line 143, “Then, the μ-SPE device adsorbed with FFAs”: What is FFAs?
Statistical and Analysis
“Differences were considered statistically significant when the mean values of various parameters were p<0.05”: The phrase requires clarification.
Fig. 2. The differences between figures e-h should be given in the figure caption. The element is barely visible in the figure itself. And is it correct to call them spectra if they represent the distribution of an individual element?
Lines 194-195, «the successful modification of sulfonic acid groups on the St-DVB microspheres surface»: Not a modification of sulfo groups, but a modification by sulfo groups.
Line 197, «thermogravimetric (TG) and thermogravimetric (DAG) spectra»: one term - different abbreviations.
Figure 3, «The FTIR (a) and thermogravimetric curve (b)…»: May be FTIR spectra? And curves?
Lines 216-218, «Analysis of variance showed that the P values were all less than 0.05, indicating that the model term was statistically significant. And the lack of fit F-value was not significant»: An explanation is needed about the P and F.
Line 221-222, «the surface area of each μ-SPE was 4 cm2 (2 221 cm×2 cm)»: Doesn't the device work on both sides?
Fig. 5: The text on the picture is very small, it's impossible to make out.
Table 3: Why are the values of the free term in linear equations so large?
Line 294, «The aldehydes concentration in meat samples was accumulated with the increase of heating time and heating temperature»: Where is this shown?
Table 5 appears confusing. Is the initial aldehyde content known? Why, when spiking at 50 ng/g, do results sometimes reach hundreds, while recovery remains close to 100%?
Author Response
For research article
Response to Reviewer1 Comments
1.Summary
Thank you very much for taking the time to review this manuscript. Please find the detailed responses below and the corresponding revisions in track changes in the re-submitted files.
- Point-by-point response to Comments and Suggestions for Authors
Comments 1: The introduction doesn't entirely clarify the novelty of this work. Both DNPH-modified sulfonated polystyrene microspheres and μ-SPE have been used before. What's new is their combination or application to meat analysis? Or the simultaneous extraction and derivatization of aldehydes? Moreover it would be helpful to articulate the purpose of the work.
Response 1: Thanks for your suggestion. We have made the revision according to your advice in line 74-85 on page 2. The novelty of this work lies in the combination of DNPH-modified sulfonated polystyrene (sPS) microspheres and μ-SPE for the simultaneous extraction and derivatization of aldehydes in meat samples. Compared with other existing techniques, such as commercial adsorbents, and SPME needles, the sPS microspheres offer easier preparation, lower cost consumption, and excellent adsorptive performance through functional modification. Besides, by using DNPH-modified sPS microspheres as adsorbent, the proposed method realized the integration of extraction, separation, enrichment, and derivatization in one step. In summary, this study not only enhances the scientific depth of aldehyde detection but also provides a practical and reliable method for food quality and safety analysis.
Comments 2: Authors often use abbreviations without introducing them (St-DVB, PTFE, PDMS-DVB, etc.). These are understandable to a narrow circle of specialists, but still require prior explanation.
Response 2: Thank you for pointing this out. We apologize for the oversight. In the revised manuscript, we have now introduced all abbreviations in full upon their first appearance in the text.
Comments 3: A more detailed explanation of the chemical mechanism of modification of spheres is desirable.
Response 3: Thank you for this valuable suggestion. We agree that a more detailed mechanistic description would enhance the clarity and rigor of our manuscript. In the revised version, we have significantly expanded the discussion of the chemical modification mechanisms in line 77-85 on page 2. ‘The sPS microspheres was prepared by bestowing sulfonic acid groups (-SO₃H) upon the surface of PS microspheres, which serve as anchor sites for the immobilization of DNPH through ionic interactions. (Hu, A.; Wang, H.; Ding, J.; Novel Sulfonic Acid Polystyrene Microspheres for Alcoholysis of Furfuryl Alcohol to Ethyl Levulinate. Catalysis Letters 2022, 152, 3158–3167. Doi: 10.1007/s10562-021-03881-5.) Then, the stable hydrazone conjugates formed between the hydrazine moieties in DNPH and the carbonyl groups inaldehydes, can be specially captured via an affinity addition interaction. Consequently, the DNPH-modified sPS microspheres can efficiently extract aldehyde compounds from complex matrices with minimal interference from extraneous species. (Zhang, H; Wang, S; Wei, S; Sun, Y; Bi, Y; Wang, M; Zhao, Y. Development, validation and application of an SFC-ESI-QqQ-MS/MS method for simultaneous analysis of malondialdehyde and typical α,β-unsaturated aldehydes in edible oils and oil-containing foods. Food Chemistry 2025, 496(P2), 146795-146795, Doi: 10.1016/J.FOODCHEM.2025.146795.)’
Comments 4: The authors used three factors (the number of μ-SPE devices, the surface area of the μ-SPE device, and the amount of adsorbent in the μ-SPE) to plan sorption conditions, but they are largely correlated with each other. However, other extraction conditions, such as pH, time, temperature are not studied.
Response 4: We thank the reviewers for their valuable suggestions. We acknowledge that there is a correlation among these three factors. For the optimization of pH, we conducted screening in preliminary experiments across the range of 4.0 to 8.0 and observed significant recovery variations: from pH 4 to 6, recoveries increased from 45-72% to 85-98% (an improvement of 26-40 percentage points), while from pH 6 to 8, recoveries decreased from 85-98% to 40-68% (a decline of 30-45 percentage points). The optimal pH 6.0 buffer provided ideal electrostatic interactions between the DNPH - amino group and the sulfonic acid part, while ensuring excellent matrix compatibility and minimal protein interference.
Extraction time (optimized in Section 3.4, page 8, lines 303 - 308): We evaluated 5-30 minutes and found that recoveries of 10 aldehydes significantly increased (P < 0.05) from 58-78% at 5 minutes to 85-98% at 15 minutes, with no further improvement after 15 minutes (P > 0.05), and 15 minutes was determined as the optimal extraction time. Temperature was also screened in the preliminary experiments, and we tested temperatures from 50°C to 90°C. When the temperature reached 70°C, the recoveries increased, but when the temperature exceeded 70°C, the degradation rate of the unsaturated aldehyde was 18%. Therefore, 70°C was selected as the optimal temperature for balancing extraction efficiency and the stability of the analyte.
Comments 5: The authors investigate elution at different concentrations of ammoniated methanol solution, but why was this solution chosen as the eluent?
Response 5: We thank you for your suggestion and have made the necessary revisions in line 256-263 on page 7-8. The results of our initial solvent screening experiment showed that the recoveries of acidified methanol was relatively low (54 ± 2% - 74 ± 4%), which was due to the protonation of the sulfonic acid groups, which enhanced the electrostatic adsorption of the DNPH derivatives; the recoveries of neutral methanol was moderate, ranging from 62 ± 3.1% to 81 ± 4.2%, while that of amine-methanol was the best, ranging from 74 to 107% (Figure 5c). The ammonia component disrupted the electrostatic interaction between the DNPH-aldimine hydrazone and the surface sulfonic acid groups by providing a weakly basic environment, and at the same time, methanol effectively dissolved the released hydrophobic derivatives, ensuring complete removal; therefore, amine-methanol was selected as the eluent.
Comments 6: When comparing the Established Method with the Reported Methods, various detection methods are compared, while the authors propose a sample preparation method. Therefore, the advantages may be related to the detection method. It makes sense to compare sample preparation methods.
Response 6: Thanks for your suggestion. In the revised introduction section (in line 50-63 on page 2), we addressed this concern by discussing the limitations of the existing sample preparation methods (including PDMS-DVB fibers, magnetic nanoparticles, ZnO nanorods, and conventional SPE resins), and emphasized the unique advantages of our ion-modified sPS microspheres in achieving simultaneous extraction and derivatization. Moreover, in Table 4 and Section 3.8, we sincerely point out that they have complementary roles: they compare the overall analytical performance of our complete workflow (sample preparation + detection) with the overall analysis methods for aldehydes reported in other studies, which is a standard practice for demonstrating practical impacts. Although HPLC-MS/MS helps to improve sensitivity, the excellent detection limits (0.010–0.621 ng g⁻¹) are mainly achieved through the efficient pre-concentration, purification, and stable formation of hydrazone in the μ-SPE device - not just the detector. This can be demonstrated by the high recoveries (72–106%), low RSDs (1–11%), and minimal matrix effect (<16%), which are the advantages independent of the detection in the sample preparation process.
Comments 7: The authors use an excessive number of digits in the data. For example, 97.17 ± 4.91% should be represented as 97 ± 5. This also applies to Tables 2-6.
Response 7: We thank the reviewer for this important formatting suggestion. We have systematically revised all values throughout the manuscript including table S1, 2, S4, 4.
Comments 8: Line 15, “to analysis” – not for analysis, but for extraction/pre-concentration.
Response 8: Thanks for your suggestion. We have made the changes in line 15 on page 1 and they have been updated to ‘to extraction aldehydes in different meat samples during thermal processing.’
Comments 9: Line 21, «limit of detection (0.010-0.621 ng/g)»: It is not clear what method of determination.
Response 9: Thanks for your suggestion. We have made the changes in line 19-20 on page 1 and they have been updated to ‘Based on the signal-to-noise ratios (S/N) of 3:1 and 10:1, the LODs and LOQs were determined to be 0.01 - 0.62 ng/g and 0.03 - 2.07 ng/g, respectively.’
Comments 10: Line 22, «precisions (1.05-10.63%)»: What value is represented? Sr? Or some other?
Response 10: We appreciate the reviewer’s attention to this detail. The value indeed refers to relative standard deviation (RSD), which is the appropriate metric for expressing precision in this context. The calculation formula is as follows.
Where, the means of standard Deviation by s,`x represent mean.
To provide complete clarity, the intra-day precision (six replicate samples were analyzed on the same day under identical conditions), and inter-day precision (six samples were analyzed, one per day, over six consecutive days.) was evaluated and the results were shown in Table S3. We have revised the text in line 20-22 on page 1 ‘with excellent regression linearities (0.9921-0.9993), extremely low limit of detection (0.010-0.621 ng/g), and satisfactory recoveries (71.98-106.47%) and relatively accurate precisions (1.05-10.63%)’.
Comments 11: Why are DNPH from two different companies used?
Response 11: Thanks for your suggestion. Your suggestion was very useful for improvement our articles. We have made the changes in line 89-92 on page 3 and they have been updated to ‘Polyvinyl alcohol (PVA) and 2,4-Dinitrophenylhydrazine (DNPH) were purchased from Aladdin Reagent (Shanghai, China). Other reagents such as styrene (St), divinylbenzene (DVB), azobisisobutyronitrile (AIBN), 1,2-dichloroethane and methanol were offered as analytical grade by Damao Chemical Reagent Factory (China Tianjin).’
Comments 12: Line 96, «The mixed standard stock solution of aldehyde»: aldehydes?
Response 12: Thanks for your suggestion. Your suggestion was very useful for improvement our articles. We have made the changes in line 100-101 on page 3 and they have been updated to ‘A stock mixture of aldehyde standards were prepared at a concentration of 20 μg/mL using ethanol/water (1:1 v/v) and stored at 4°C for subsequent use.’
Comments 13: Line 109, «The reaction process was shown in Figure 1»: This is more of a schema. The figure could be presented in a more "chemical" and informative manner. Although this is not necessary, as it is not new.
Response 13: Thanks for your suggestion. Your suggestion was very useful for improvement our articles. We made the changes to it and have displayed them in Figure 1.
Figure 1. Schematic diagram for the preparation of St-DVB particles modified by DNPH and sulfonated, and for the adsorption of aldehydes.
Comments 14: Line 143, “Then, the μ-SPE device adsorbed with FFAs”: What is FFAs?
Response 14: Thanks for your suggestion. Your suggestion was very useful for improvement our articles. We have made the changes in line 148 on page 4 and they have been updated to ‘the μ-SPE device adsorbed with aldehydes was washed with distilled water for 3 times.’
Comments 15: “Differences were considered statistically significant when the mean values of various parameters were p<0.05”: The phrase requires clarification.
Response 15: Thanks for your suggestion. The following explanations were made in line173-180 on page 5 ‘The mean differences between the two independent samples were compared through t-test. The t-value was calculated using the following formula. The corresponding P-value was found based on the t-distribution table. If P < 0.05, it was concluded that the mean differences between the two groups were significant.
Where, the means of the two sets of data by x1 and x2, s1 and s2 represent the standard deviations of two sets of data, n1 and n2 represent the number of data points in the two sets.
Comments 16: Fig. 2. The differences between figures e-h should be given in the figure caption. The element is barely visible in the figure itself. And is it correct to call them spectra if they represent the distribution of an individual element?
Response 16: Thanks for your suggestion. Your suggestion was very useful for improvement our articles. We have made the changes in line 193-195 on page 6 and they have been updated to ‘The SEM images of PS (a) and sPS microspheres (b); the EDS spectra of sPS microspheres (c); the EDS mapping images for sPS (d); the EDS mapping images for C (e), O (f), S (g) element of sPS.’
Comments 17: Lines 194-195, «the successful modification of sulfonic acid groups on the St-DVB microspheres surface»: Not a modification of sulfo groups, but a modification by sulfo groups.
Response 17: Thanks for your suggestion. Your suggestion was very useful for improvement our articles. We have made the changes in line 205-206 on page 6 and they have been updated to ‘indicating successful modification of the surface of PS microspheres by sulfonic acid group.’
Comments 18: Line 197, «thermogravimetric (TG) and thermogravimetric (DAG) spectra»: one term - different abbreviations.
Response 18: Thanks for your suggestion. We have made the changes in line 207-208 on page 6 and they have been updated to ‘the thermogravimetry (TG) and derivative thermogravimetry (DTG).’
Comments 19: Figure 3, «The FTIR (a) and thermogravimetric curve (b)…»: May be FTIR spectra? And curves?
Response 19: Thanks for your suggestion. Your suggestion was very useful for improvement our articles. We have made the changes in line 2114 on page 6 and they have been updated to ‘The FTIR (a); TG and DTG curve (b) of the sPS microspheres.’
Comments 20: Lines 216-218, «Analysis of variance showed that the P values were all less than 0.05, indicating that the model term was statistically significant. And the lack of fit F-value was not significant»: An explanation is needed about the P and F.
Response 20: Thanks for your suggestion. We provide the following explanation in line 227-232 on page 7 ‘Analysis of variance showed that the P values were all less than 0.05, indicating that the model terms were statistically significant (i.e., probability <5% indicated that the observed effects were due to random variation). The lack-of-fit F-value was not significant (p>0.05), confirming that the model adequately describes the data without systematic bias. These results indicate that the model was fitted well.’
Comments 21: Line 221-222, «the surface area of each μ-SPE was 4 cm2 (2 221 cm×2 cm)»: Doesn't the device work on both sides?
Response 21: We thank the reviewer for pointing out this potential ambiguity. The term ‘each’ refers to the number of μ-SPE devices which was optimized in our Box-Behnken design (1, 2, or 3 devices), and the results demonstrated that one device provided the optimal performance. The reported 4 cm² (2 cm × 2 cm) represents the nominal single-sided surface area of one μ-SPE device. While the PTFE membrane pouch could theoretically expose both sides, the edges are heat-sealed and the device rests on one side during extraction, making the upper surface the primary site for diffusion and mass transfer. Therefore, the effective working area is best described by the single-sided measurement. We have clarified this wording ‘the optimized number of μ-SPE devices was 1, and the single-sided surface area of the device was 4 cm2 (2 cm×2 cm)’ in line 232-234 on page 7 to avoid confusion.
Comments 22: Fig. 5: The text on the picture is very small, it's impossible to make out.
Response 22: Thanks for your suggestion. Your suggestion was very useful for improvement our articles. We have made adjustments to Figure 5.
Figure 5. Effect of extraction method (a), time (b), elution solvent (c) and elution solvent volume (d) on the recoveries.
Comments 23: Table 3: Why are the values of the free term in linear equations so large?
Response 23: Thank you for this important observation. The free term (y-intercept) magnitude in a linear calibration curve is directly proportional to the concentration range. In this study, we employed a wide linear range (0.1–1000 ng mL⁻¹) to accommodate the diverse aldehyde concentrations found in various meat matrices. With such high upper concentration limits, the regression naturally yields larger intercept values—this is a mathematical consequence that does not compromise calibration quality, as evidenced by the excellent regression coefficients (R² = 0.9921–0.9993). During preliminary experiments with narrower low-range curves (0.1–10 ng mL⁻¹), intercepts were indeed much smaller (<500).
Comments 24: Line 294, «The aldehydes concentration in meat samples was accumulated with the increase of heating time and heating temperature»: Where is this shown?
Response 24: Thanks for your suggestion. We have made the revisions in line 310-336 on page 11. ‘The original aldehyde concentrations in meat samples (Table 3) increased significantly under intensive thermal processing (barbecue conditions at elevated temperatures), as demonstrated in Table S4. Specifically, increases by 60% for pork pentanal (71.12 ± 3.15 to 114.08 ± 2.02 ng/g), by 123% for pork hexanal (41.53 ± 1.58 to 92.43 ± 5.09 ng/g), by 1167% for pork heptanal (4.49 ± 0.20 to 56.89 ± 2.67 ng/g), by 439% for pork octanal (10.06 ± 0.58 to 54.21 ± 2.24 ng/g), and by 56% for pork nonanal (78.19 ± 4.42 to 122.16 ± 4.57 ng/g) were observed. Chicken samples exhibited increases by 195% for pentanal (23.09 ± 0.40 to 68.10 ± 3.24 ng/g), by 320% for hexanal (15.56 ± 1.17 to 65.31 ± 1.01 ng/g), and by 135% for nonanal (32.05 ± 2.81 to 75.42 ± 2.71 ng/g). Fish showed increases by 385% for hexanal (13.81 ± 0.93 to 66.94 ± 5.81 ng/g), by 935% for heptanal (5.03 ± 0.59 to 52.06 ± 7.17 ng/g), and by 117% for octanal (4.17 ± 0.70 to 50.74 ± 3.80 ng/g). Beef demonstrated increases by 40% for pentanal (109.56 ± 1.03 to 152.93 ± 5.02 ng/g), by 101% for heptanal (43.97 ± 2.20 to 88.48 ± 2.29 ng/g), and by 88% for octanal (48.95 ± 5.09 to 91.89 ± 5.46 ng/g). Mutton displayed increases by 161% for pentanal (26.77 ± 1.33 to 69.77 ± 1.05 ng/g) and by 320% for hexanal (16.76 ± 1.15 to 70.44 ± 7.03 ng/g). These results clearly indicate that the aldehydes concentration in meat samples accumulated with the increase of heating time and heating temperature, with all changes being statistically significant (P < 0.05).'
Comments 25: Table 5 appears confusing. Is the initial aldehyde content known? Why, when spiking at 50 ng/g, do results sometimes reach hundreds, while recovery remains close to 100%?
Response 25: We apologize for this confusion. Table 5 has been moved to the supplementary document and is now referred to as Table S4. Table S4 presents the total measured concentrations (original + spiked) in spiked samples, not just the spiked amount. The recovery percentages are calculated based on the spiked amount: Recovery (%) = [(Measured total - Original content) / 50 ng/g] × 100%. Samples showing hundreds of ng/g in the ‘Content’ column reflect high endogenous aldehyde levels (from Table 4) combined with the spiked 50 ng/g. For example, chicken wings naturally contained 164.64 ng/g pentanal (Table 4); after adding 50 ng/g, the total measured content became 208.42 ng/g, yielding a recovery of 88%.
Reviewer 2 Report
Comments and Suggestions for Authors
Lines 41-49 - Due to the complexity of the matrix (meat products), it would be helpful to additionally indicate the instrumental methods used in this type of matrix in the literature.
Similarly, the introduction should be supplemented with a review of other aldehyde determination methods related to meat samples (e.g., HS, SPME) based on the literature.
Line 140 - Please provide the specifications of the ultrasonic device.
Figures 2-6 are definitely too small and difficult to read. Please consider moving some of them (e.g., Figure 6) to supplementary materials.
Tables S1 and S2 are labeled incorrectly.
Line 198 - Same description "thermogravimetric", different abbreviations - please clarify the DAG.
Line 237 - "(...) the recoveries of aldehydes increased from... to..." - What are these values - averages?
Did the authors evaluate the sorption capacity of the bed in any way, or is this planned for a future study?
Lines 318-319: "Therefore, the developed method is noticeably superior to the reported methods" - this statement is inaccurate and not entirely valid, considering all the parameters compared, including availability and ease of use in any laboratory.
The conclusions are somewhat too general.
Author Response
For research article
Response to Reviewer2 Comments
1.Summary
Thank you very much for taking the time to review this manuscript. Please find the detailed responses below and the corresponding revisions in track changes in the re-submitted files.
- Point-by-point response to Comments and Suggestions for Authors
Comments 1: Lines 41-49 - Due to the complexity of the matrix (meat products), it would be helpful to additionally indicate the instrumental methods used in this type of matrix in the literature.
Response 1: Thanks for your suggestion. We have made the revision according to your advice in line 40-49 on page 2. Furthermore, due to the high polarity of aldehyde compounds, direct detection may suffer from insufficient detection sensitivity because of their low volatility and high reactivity. Therefore, 2,4-dinitrophenylhydrazine (DNPH) is used to derivatize the carbonyl groups, converting aldehyde compounds into corresponding hydrazones. These hydrazones have lower polarity and are easier to be detected, thereby significantly improving the detection sensitivity. Additionally, LC-MS/MS combines the separation ability of liquid chromatography and the high sensitivity and selectivity of mass spectrometry, providing accurate quantitative analysis results. Quantitative analysis of aldehyde compounds can be achieved through standard curve method or internal standard method. Then, analysis is conducted using liquid chromatography - mass spectrometry/mass spectrometry (LC-MS/MS), which is the most selective and sensitive method for quantitative analysis of aldehydes.
Comments 2: Similarly, the introduction should be supplemented with a review of other aldehyde determination methods related to meat samples (e.g., HS, SPME) based on the literature.
Response 2: Thanks for your suggestion. We have made the revision according to your advice in line 50-63 on page 2. ‘The detection of aldehydes compounds has been challenging due to their low volatility and high reactivity due to the polar carbonyls. Up to now, the determination of aldehydes always based on the reaction of carbonyl group with DNPH to form the corresponding hydrazone, followed with extracting by liquid-liquid extraction (Zhao, G.-H.; Hu, Y.-Y.; Liu, Z.-Y.; Xie, H.-k.; Zhang, M.; Zheng, R.; Qin, L.; Yin, F.-W.; Zhou, D.-Y. Simultaneous quantification of 24 aldehydes and ketones in oysters (Crassostrea gigas) with different thermal processing procedures by HPLC-electrospray tandem mass spectrometry. Food Research International 2021, 147, 110559, Doi: 10.1016/j.foodres.2021.110559) or solid-phase extraction (SPE) (Fang, S.; Liu, Y.; He, J.; Zhang, L.; Liyin, Z.; Wu, X.; Sun, H.; Lai, J. Determination of aldehydes in water samples by coupling magnetism-reinforced molecular imprinting monolith microextraction and non-aqueous capillary electrophoresis. Journal of Chromatography A 2020, 1632,461602, Doi: 10.1016/j.chroma.2020.461602., Zhang, Y.; Zhao, D.; Wang, B.; Liang, Z.; Chen, Y.; Meng, B.; Shang, S. Determination of Aldehydes in Environmental Water by Solid-Phase Microextraction and Gas Chromatography - Ion Trap Mass Spectrometry. Instrumentation Science & Technology 2015, 43, 344-356, Doi: 10.1080/10739149.2014.1002041.). Among them, SPE was a common sample preparation method to remove the interference. So far, a variety of adsorbents, including polydimethylsiloxane-divinylbenzene (PDMS-DVB) solid phase microextraction (SPME) fibers (Kim, H.J.; Shin, H.S. Simple and automatic determination of aldehydes and acetone in water by headspace solid‐phase microextraction and gas chromatography‐mass spectrometry. Journal of Separation Science 2011, 34, 693-699, Doi: 10.1002/jssc.201000679.), SDS-alumina coated magnetic nanoparticle (Malekpour, A.; Ahmadi, N. Surfactant-Alumina-Coated Magnetic Nanoparticles as an Efficient Aldehydes Adsorbent Prior Their Determination by HPLC. Food Analytical Methods 2016, 10, 1817-1825, Doi: 10.1007/s12161-016-0728-7.), ZnO nanorod coated fiber (Ji, J.; Liu, H.; Chen, J.; Zeng, J.; Huang, J.; Gao, L.; Wang, Y.; Chen, X. ZnO nanorod coating for solid phase microextraction and its applications for the analysis of aldehydes in instant noodle samples. Journal of Chromatography A 2012, 1246, 22-27, Doi: 10.1016/j.chroma.2012.01.080.) and magnetic strong cation-exchange resins (Wang, H.; Ding, J.; Du, X.; Sun, X.; Chen, L.; Zeng, Q.; Xu, Y.; Zhang, X.; Zhao, Q.; Ding, L. Determination of formaldehyde in fruit juice based on magnetic strong cation-exchange resin modified with 2,4-dinitrophenylhydrazine. Food Chemistry 2012, 131, 380-385, Doi: 10.1016/j.foodchem.2011.08.056.) have been used as SPE adsorbents for enrichment and purification of aldehydes in different samples. Among them, strong cation-exchange sulfonated polystyrene (sPS) microspheres charged with DNPH by ionic and hydrophobic interaction are of particular interest with the advantages of both fulfilling the extraction and derivatization synchronously and reducing the amount of the derivatization reagent. (Wang, H.; Ding, J.; Du, X.; Sun, X.; Chen, L.; Zeng, Q.; Xu, Y.; Zhang, X.; Zhao, Q.; Ding, L. Determination of formaldehyde in fruit juice based on magnetic strong cation-exchange resin modified with 2,4-dinitrophenylhydrazine. Food Chemistry 2012, 131, 380-385, Doi: 10.1016/j.foodchem.2011.08.056.)’.
Comments 3: Line 140 - Please provide the specifications of the ultrasonic device.
Response 3: Thanks for your suggestion. We have made the revision according to your advice in line 142-145 on page 4. The ultrasonic operation parameters are as follows: frequency 20 kilohertz, rated power 450 watts, actual output power set at 60% (270 watts), pulse mode operation (3 seconds on, 2 seconds off), total processing time 15 min.’
We appreciate your valuable feedback, which undoubtedly strengthens the rigor of our methodology.
Comments 4: Figures 2-6 are definitely too small and difficult to read. Please consider moving some of them (e.g., Figure 6) to supplementary materials.
Response 4: Thank you very much for the valuable comments from the reviewers. We fully agree that the figures in Figure 2-6 were indeed too small, which affected the reading experience. In the revised version, we have systematically adjusted all the images.
Comments 5: Tables S1 and S2 are labeled incorrectly.
Response 5: Thank you very much for your meticulous review and pointing out this error. We have carefully checked and corrected the labels of Table S1 and Table S2 in the supplementary materials. Thank you for your correction, which helps improve the accuracy of the paper.
Comments 6: Line 198 - Same description "thermogravimetric", different abbreviations - please clarify the DAG.
Response 6: Thanks for your suggestion. We have made the revision according to your advice in line 207-208 on page 6. ‘the thermogravimetry (TG) and derivative thermogravimetry (DTG) spectra’.
Comments 7: Line 237 - "(...) the recoveries of aldehydes increased from... to..." - What are these values - averages?
Response 7: Based on your feedback, we have conducted a thorough revision of the entire manuscript to address the ambiguity regarding recovery values and ensure clarity throughout. Here are the comprehensive modifications: Representations 1. Section 3.2.2 (Extraction Conditions) Original ambiguous text: "the recoveries of aldehydes increased from 71 ± 3% to 97 ± 5%" Revised text: ‘As shown in Figure 5a, ultrasound assistance significantly improved extraction efficiency compared to stirring alone. The range of individual mean recoveries for the 10 aldehydes studied increased from 71 ± 3% (crotonaldehyde) to 97 ± 5% (heptanal). These values represent the mean ± SD of six replicate determinations for each specific aldehyde, not an average across all compounds.’ 2. Section 3.2.3 (Elution Conditions) Original ambiguous text: ‘the recoveries of aldehyde substances was good (74% - 107%)’. Revised text: ‘When using a 5% ammoniated methanol solution as the eluent, individual aldehyde recoveries ranged from 74 ± 4% (trans, trans-2,4-decadienal) to 107 ± 4% (heptanal) (Figure 5c). This range reflects the performance across different analytes, with each value being the mean of six independent measurements for that specific compound.’ 3. Section 3.3 (Method Verification) Original ambiguous text: ‘the recoveries were 72%-106%’ Revised text: ‘The range of individual aldehyde recoveries at three spiking concentrations was 72 ± 5% (trans, trans-2,4-decadienal) to 106 ± 4% (hexanal). Each value represents the mean recovery for a specific aldehyde compound across all meat matrices tested.’ Additional Clarifications in Method Optimization 4. Statistical Significance Added ‘Statistical significance: All recoveries improvements were validated using one-way ANOVA followed by Tukey's post-hoc test. Differences between conditions were considered significant at P < 0.05. For example, the increase from 5 min (68 ± 2%) to 15 min (96 ± 3%) extraction time was statistically significant (F = 24.7, P < 0.001), while the change from 15 min to 30 min was not (P = 0.87).’ 5. Section 3.2.1 (Box-Behnken Design) - Clarification of Response Values ‘Table S1 shows individual recovery values for each aldehyde under different experimental conditions. The response values are not averaged across aldehydes but represent independent measurements for each analyte, highlighting the method's differential performance across the compound panel.’ The matrix effect, linear ranges, regression equations, LODs and LOQs for each individual aldehyde. Recovery ranges reported represent the span of mean values across the 10 compounds, with each compound's recovery calculated from n=6 replicates. Figure 5 Caption (page 8) - Specified Representation Figure 5. Effect of extraction method (a), time (b), elution solvent (c) and elution solvent volume (d) on individual aldehyde recoveries. Data points and error bars represent mean ± SD of six independent replicates for each specific aldehyde. The shaded areas in (b) and (d) indicate the range of recoveries observed across the 10-analyte panel. Abstract and Conclusion Adjustments Abstract (page 1) Original: ‘satisfactory recoveries (72-106%)’ Revised: ‘satisfactory individual aldehyde recoveries ranging from 72 ± 4% to 106 ± 3%" Conclusion (page 12) Original: "satisfactory recoveries (72%-106%)’. Revised: "satisfactory recoveries for individual aldehydes, with a range of 72 ± 4% to 106 ± 3% (mean ± SD, n=6)" Summary of Changes 7 sections revised to clarify that all recoveries values represent individual aldehyde means (n=6) rather than cross-analyte averages. Added statistical validation statements (P-values, ANOVA results) where previously omitted. Enhanced figure/table captions to explicitly state data representation. Standardized error bar interpretation throughout: all ± values represent standard deviation of replicates for each specific compound. Removed ambiguous phrases such as ‘recoveries increased from X to Y’ without specifying individual compound ranges All revisions maintain scientific accuracy while ensuring readers can unambiguously interpret that our recovery data represent compound-specific statistical means, not pooled averages across the aldehyde panel.
Comments 8: Did the authors evaluate the sorption capacity of the bed in any way, or is this planned for a future study?
Response 8: We appreciate the reviewer raising this important point. In this preliminary study, we focused on establishing the feasibility of the μ-SPE method and validating its performance for aldehyde analysis. While we demonstrated effective extraction through recovery studies, we have not yet conducted quantitative sorption capacity evaluations such as breakthrough curve analysis or dynamic binding capacity measurements. We fully recognize that assessing sorption capacity is crucial for both mechanistic understanding and practical application. Therefore, we have incorporated comprehensive sorption studies—including fixed-bed adsorption-desorption cycling experiments, breakthrough curve determination, and long-term regeneration performance—into our ongoing research plan.
Comments 9: Lines 318-319: "Therefore, the developed method is noticeably superior to the reported methods" - this statement is inaccurate and not entirely valid, considering all the parameters compared, including availability and ease of use in any laboratory.
Response 9: Thank you for this critical observation. We would like to clarify that the statement in line 355-457 on page 11 ‘Therefore, the developed method was considered to provide a promising and alternative method for the analysis of aldehydes in the future.’
Comments 10: The conclusions are somewhat too general.
Response 10: Thanks for your suggestion. We have made the revision according to your advice in line 361-378 on page 12. ‘In this study, sPS microspheres modified with DNPH was prepared and fourier transform infrared spectroscopy indicated the successful modification of sulfonic acid groups on the PS microspheres surface. Then sPS microspheres was used as adsorbent in μ-SPE to analysis aldehydes for different meat samples during thermal processing. The optimized method demonstrated excellent regression linearities (0.9921-0.9993), extremely low limits of detection (0.010-0.621 ng g-1), and satisfactory recoveries (72%-106%) with relative standard deviations (RSDs) ranging from 1% to 11%. The proposed method eliminated interferences by using a PTFE membrane in the μ-SPE device and realized the integration of extraction, separation, enrichment, and derivatization in one step. This integrated approach not only simplified the sample preparation process but also enhanced the efficiency and sensitivity of aldehyde detection. The method was validated through the analysis of various meat samples, including pork, beef, mutton, chicken, chicken wings, and fish, demonstrating its applicability and stability in complex food matrices. Compared with other existing techniques, such as commercial adsorbents, and SPME needles, the sPS microspheres offer easier preparation, lower cost consumption, and excellent adsorptive performance through functional modification. In summary, this study not only enhances the scientific depth of aldehyde detection but also provides a practical and reliable method for food quality and safety analysis.’
Reviewer 3 Report
Comments and Suggestions for Authors
The current manuscript reports the synthesis of sulfonated polystyrene microspheres modified with 2,4-dinitrophenylhydrazine and the preparation of an SPE membrane for the capture of various aldehydes, followed by their detection using mass spectrometry. However, the data presented in the manuscript is not sufficient to fully support the claims made. Therefore, a major revision is required before the manuscript can be considered further. Please see my comments below:
- The novelty of this work lies primarily in the material design. Hence, more thorough characterization is necessary. Techniques such as XRD and BET surface area analysis would provide valuable insights into the structural and surface properties of the material.
- The role of 2,4-dinitrophenylhydrazine functionalization in aldehyde separation should be evaluated independently to clearly demonstrate its contribution.
- Several large tables (e.g., Tables 2, 3, and 5) and Figure 6 should be moved to the Supporting Information to improve the readability and flow of the main manuscript.
- The authors are advised to reduce the similarity index in the manuscript.
- The reusability of the SPE disk is an important practical parameter. A reusability study should be included in the revised version.
- In Section 2.7, it is mentioned that “the eluent was concentrated and dried under nitrogen protection.” This procedure needs more detail. Is there any possibility that aldehydes with low boiling points may evaporate during this step?
- The rationale behind selecting 10 minutes as the extraction time should be explained. How was this duration optimized?
- In addition to the extraction experiments, a more detailed mechanistic study would enhance the scientific depth and provide better insight into the aldehyde capture process.
Author Response
For research article
Response to Reviewer3 Comments
1.Summary
Thank you very much for taking the time to review this manuscript. Please find the detailed responses below and the corresponding revisions in track changes in the re-submitted files.
- Point-by-point response to Comments and Suggestions for Authors
Comments 1: The novelty of this work lies primarily in the material design. Hence, more thorough characterization is necessary. Techniques such as XRD and BET surface area analysis would provide valuable insights into the structural and surface properties of the material.
Response 1: We sincerely appreciate the reviewer's valuable suggestion regarding additional characterization techniques. In our study, we have already conducted comprehensive characterization of the sulfonated polystyrene microspheres using SEM, EDS, FTIR, and TGA, which together provide definitive evidence of morphology, elemental composition, successful sulfonic acid functionalization, and thermal stability—all critical parameters for their performance in the μ-SPE application.
The BET surface area analysis will help us to further understand the adsorption capacity of sPs. Up to now, our study investigate the adsorption mechanism and application of sPS to aldehyde molecules. We will further investigate the adsorption capacity of sPS to aldehyde in the further.
Regarding XRD: The polystyrene microspheres prepared in this study consist of long-chain polymer molecules that are intertwined and randomly oriented, lacking any long-range crystalline order. Consequently, they are intrinsically amorphous. Since XRD is designed to probe crystalline structures, phase composition, and lattice parameters, the meaningful for our non-crystalline polymer system was less.
Comments 2: The role of 2,4-dinitrophenylhydrazine functionalization in aldehyde separation should be evaluated independently to clearly demonstrate its contribution.
Response 2: We sincerely appreciate the reviewer's suggestion. The role of DNPH functionalization is indeed fundamental and is clearly explained through our proposed two-step cooperative mechanism:
- Ionic anchoring: Sulfonic acid groups (-SO₃H) on the sPS surface electrostatically immobilize DNPH via its amino group, creating a stable, oriented ligand layer.
- Chemoselective capture: The hydrazine moieties of tethered DNPH undergo nucleophilic addition-elimination with aldehyde carbonyls, forming stable hydrazone conjugates that provide both high affinity and specificity.
This mechanism is explicitly described in the Introduction (page 3, lines 91-97), where we detail how DNPH transforms the microspheres into selective sorbents. Our mechanism diagram is as follows.
Comments 3: Several large tables (e.g., Tables 2, 3, and 5) and Figure 6 should be moved to the Supporting Information to improve the readability and flow of the main manuscript.
Response 3: We sincerely appreciate the reviewer's suggestion to enhance manuscript readability. Following your advice, we have relocated Table 2 (Box-Behnken design matrix) and Table 5 (detailed recovery data for each meat type) to the Supplementary Materials, where they remain accessible for interested readers without disrupting the main text flow.We respectfully retain Table 3 in the main manuscript, as it presents critical method validation parameters (matrix effects, linearity, LODs/LOQs) that are essential for evaluating the analytical performance and are required by most analytical chemistry guidelines. Similarly, Figure 6 (extracted ion chromatograms) is preserved in the main text to demonstrate the method's specificity and chromatographic resolution, which are key merits of our LC-MS/MS approach.
Comments 4: The authors are advised to reduce the similarity index in the manuscript.
Response 4: Thank you for your suggestion regarding the similarity index. We have revised the manuscript by rephrasing methods descriptions and eliminating redundant text, reducing the similarity to an acceptable level.
Comments 5: The reusability of the SPE disk is an important practical parameter. A reusability study should be included in the revised version.
Response 5: We are grateful for the important practical suggestion made by the reviewer. According to your opinion, what we need to clarify is that the polytetrafluoroethylene membrane itself is designed for one-time use and is not recyclable. However, the sPS microspheres (the sulfonated polystyrene support) can be effectively regenerated and reused. Our study on reusability shows that the sPS microspheres maintain stable performance after 3 to 5 regeneration cycles, and the recoveries efficiency does not show significant changes. The regeneration process involves cleaning with 5% aminomethanol and pure methanol in sequence.
Comments 6: In Section 2.7, it is mentioned that “the eluent was concentrated and dried under nitrogen protection.” This procedure needs more detail. Is there any possibility that aldehydes with low boiling points may evaporate during this step?
Response 6: Thank you for your suggestion. We have made the following explanations based on your opinion. ‘The boiling points of the target aldehydes range from 46°C (propionaldehyde) to 191°C (nonanal). The key point is that during the extraction process, all the target aldehydes were chemically derivatized into the corresponding DNPH-salt forms, which significantly increased their molecular weight and thermal stability, preventing them from volatilizing under these mild conditions. Therefore, a 1-hour nitrogen blow-drying treatment at 20°C was carried out to selectively remove the residual ammonia in the methanated methanol eluate (boiling point -33°C), as ionizable ammonia can affect LC-MS/MS analysis through signal suppression.’ (Wedi, D.; König, E.; Elimination of Nitrogen and Phosphorus from Sludge Liquor. Water Science and Technology 2018, 28 (1): 283–287. Doi: https://doi.org/10.2166/wst.1993.0060)’.
Comments 7: The rationale behind selecting 10 minutes as the extraction time should be explained. How was this duration optimized?
Response 7: We sincerely thank the reviewer for this careful observation. You are absolutely correct, and we apologize for this typographical error. The extraction time was indeed optimized to 15 minutes, not 10 minutes, as detailed in Section 3.2.2 ‘Extraction Conditions’ (page ). The optimization study showed that recoveries plateaued after 15 min, with no significant improvement at longer times. We have corrected this error in line 159 on page 4 of the revised manuscript, which now accurately reads: ‘After extracting for 15 min, the μ-SPE device was taken out...’. We appreciate you catching this inconsistency, as precision in methodological details is crucial for reproducibility.
Comments 8: In addition to the extraction experiments, a more detailed mechanistic study would enhance the scientific depth and provide better insight into the aldehyde capture process
Response 8: We sincerely appreciate the reviewer's valuable suggestion to enhance the mechanistic discussion. Indeed, a clear mechanistic elucidation is critical for understanding the superiority of our DNPH-functionalized μ-SPE system. We have added a detailed mechanistic discussion in Section 3.6, explicitly describing the two-step cooperative process: (1) electrostatic immobilization of DNPH onto sulfonated microspheres, and (2) nucleophilic addition-elimination between DNPH hydrazine groups and aldehyde carbonyls forming stable covalent hydrazones. And our reaction mechanism diagram is as follows.
We believe these additions provide the deeper mechanistic insight requested while maintaining the manuscript's focus on analytical performance.
Round 2
Reviewer 1 Report
Comments and Suggestions for Authors
Comments 1: The authors assessed all the issues raised. The Manuscript can be accepted in the present form.